# Vitamin D and Autoimmune Rheumatic Diseases

**DOI:** 10.3390/biom13040709

**Published:** 2023-04-21

**Authors:** Lambros Athanassiou, Ifigenia Kostoglou-Athanassiou, Michael Koutsilieris, Yehuda Shoenfeld

**Affiliations:** 1Department of Rheumatology, Asclepeion Hospital, Voula, GR16673 Athens, Greece; 2Department of Physiology, Medical School, University of Athens, GR11527 Athens, Greece; 3Department of Endocrinology, Asclepeion Hospital, Voula, GR16673 Athens, Greece; 4Zabludowicz Center for Autoimmune Diseases, Sheba Medical Center, Tel Aviv University, Tel Aviv 69978, Israel

**Keywords:** vitamin D, rheumatoid arthritis, systemic lupus erythematosus, ankylosing spondylitis, Sjogren’s syndrome, systemic sclerosis

## Abstract

Vitamin D is a steroid hormone with potent immune-modulating properties. It has been shown to stimulate innate immunity and induce immune tolerance. Extensive research efforts have shown that vitamin D deficiency may be related to the development of autoimmune diseases. Vitamin D deficiency has been observed in patients with rheumatoid arthritis (RA) and has been shown to be inversely related to disease activity. Moreover, vitamin D deficiency may be implicated in the pathogenesis of the disease. Vitamin D deficiency has also been observed in patients with systemic lupus erythematosus (SLE). It has been found to be inversely related to disease activity and renal involvement. In addition, vitamin D receptor polymorphisms have been studied in SLE. Vitamin D levels have been studied in patients with Sjogren’s syndrome, and vitamin D deficiency may be related to neuropathy and the development of lymphoma in the context of Sjogren’s syndrome. Vitamin D deficiency has been observed in ankylosing spondylitis, psoriatic arthritis (PsA), and idiopathic inflammatory myopathies. Vitamin D deficiency has also been observed in systemic sclerosis. Vitamin D deficiency may be implicated in the pathogenesis of autoimmunity, and it may be administered to prevent autoimmune disease and reduce pain in the context of autoimmune rheumatic disorders.

## 1. Introduction

Vitamin D is a steroid hormone with potent immune-modulating properties [1,2]. Its immune-modulating properties were discovered very early on in the journey of its discovery [3,4,5]. It was observed that it might be a factor contributing to the treatment of tuberculosis [6,7,8]. Later it was shown that it might help in the treatment of leprosy [9,10]. Further data showed that vitamin D might prevent exacerbations of asthmatic attacks in patients with the chronic obstructive pulmonary disease [11]. It has also been shown that vitamin D may prevent the recurrence of urinary infections [12]. Vitamin D plays a significant, potent, and multimodal role in the regulation of the immune system [1,13]. It acts to regulate innate and adaptive immunity [14,15]. Its immunoregulatory role in the prevention and treatment of autoimmune diseases is the focus of research interest [16,17]. Vitamin D deficiency has been observed in various rheumatic autoimmune diseases, and it may be related to the pathogenesis of autoimmunity [16,18]. Moreover, vitamin D is administered in patients with rheumatic autoimmune diseases to control pain and prevent disease exacerbation [19,20,21]. Vitamin D has been proposed to prevent autoimmunity [21,22]. 

Multiple epidemiological studies from all over the world have now shown the relationship between vitamin D deficiency with autoimmune diseases. The relationship of vitamin D deficiency with rheumatoid arthritis (RA) [23,24], systemic lupus erythematosus (SLE) [25,26,27], inflammatory bowel disease [28,29], multiple sclerosis [30,31], diabetes mellitus type 1 [32], Hashimoto’s thyroiditis [33] has been shown. In particular, the relationship between vitamin D deficiency with autoimmune rheumatic diseases has attracted the attention of the scientific community and has led to various studies of vitamin D supplementation to mitigate the severity of autoimmune rheumatic diseases [21]. The aim was to present the relationship of vitamin D deficiency with autoimmune rheumatic diseases, namely, RA, SLE, ankylosing spondylitis, psoriatic arthritis (PsA), Sjogren’s syndrome, inflammatory myopathies and systemic sclerosis, and the role of vitamin D supplementation in the above-named disorders. Randomized control trials, cohort studies, and meta-analyses on the relationship between vitamin D and RA, SLE, ankylosing spondylitis, PsA, Sjogren’s syndrome, systemic sclerosis, and inflammatory myopathies performed between 2000 and 2022 were reviewed. This review will cover recent data on the multimodal immunomodulatory role of vitamin D and the relationship of vitamin D deficiency with autoimmunity. 

## 2. Vitamin D and Immunity

### 2.1. Vitamin D and Innate Immunity

The important role of vitamin D in the regulation of the innate immune system is shown in its effect at multiple levels, namely at the level of the skin [34], the intestinal epithelium [35], the airway epithelium, and cells of the innate immune system [36,37]. Vitamin D is involved in the formation of the permeability barrier in the skin, it is produced and acts within keratinocytes to enhance the production of defensin β_2_ and cathelicidin, as they harbor both the enzymatic machinery to produce 1,25(OH)_2_D_3_ and the vitamin D receptor (VDR) to respond to it [34]. It also acts to maintain the epithelial barrier function in the intestine, as it regulates tight junctions [38,39] and intestinal epithelial cell apoptosis [40]. Cells of the airway epithelium and alveolar macrophages harbor 1-hydroxylase, the enzyme responsible for the conversion of 25(OH)D_3_ to 1,25(OH)_2_D_3_ and VDR and thus produce and respond to vitamin D, enabling the response to infectious agents invading via the airway [41]. Vitamin D also acts on macrophages and monocytes to enhance the response to infectious agents such as mycobacterium tuberculosis and mycobacterium leprae [7,8,9]. The monocytes and macrophages express the VDR, although the expression of VDR decreases as monocytes differentiate from macrophages [42]. Vitamin D induces the production of defensin β_2_ and cathelicidin in response to invading infectious agents by macrophages, monocytes, and keratinocytes [14]. Cathelicidin is an antimicrobial agent with activity against gram-positive and gram-negative bacteria acting through cell lysis via cell membrane destabilization [43]. It also displays activity against viruses and fungi [44]. 25(OH)D_3_ is the major circulating form of vitamin D used to determine vitamin D status and is important for the local production of 1,25(OH)_2_D_3_, which upregulates cathelicidin production in the skin, the airway, and macrophages. The exposure of human monocytes to pathogens increases the expression of both 1,25(OH)_2_D_3_ and VDR, thus increasing both the local production of 1,25(OH)_2_D_3_ and the ability of the cell to respond to it [14]. Ultraviolet light may directly stimulate cathelicidin production by providing the substrate 25(OH)D_3_ directly from vitamin D_3_ produced within the skin [45,46]. 1,25(OH)_2_D_3_ is involved in macrophage differentiation and activation. Macrophage exposure to 1,25(OH)_2_D_3_ can induce the differentiation of macrophages from monocytes, and following exposure to inflammatory immune factors, the expression of 1-hydroxylase is enhanced, thus allowing the macrophage to locally produce 1,25(OH)_2_D_3_ [47,48], which is necessary for modulation of the immune response. These data show that vitamin D is a regulator of innate immunity [15], acting both on macrophages and monocytes as well as on keratinocytes, gut epithelial cells, airway epithelial cells, and alveolar macrophages [49]. 

Neutrophils are a white blood cell population, and they contribute to a line of defense against microbial pathogens. Neutrophils express a functional vitamin D receptor [50]. 1,25(OH)_2_D_3_ administration has been found to down-regulate neutrophil function and activity by reducing the production of inflammatory cytokines and reactive oxygen species [51]. 

### 2.2. Vitamin D and Dendritic Cells

Dendritic cells are a target of vitamin D [52]. Dendritic cells are cells of the immune system which act to survey the body for signs of invasion by pathogens. When they encounter a foreign substance, they engulf it, and after processing it, they present parts of it to T cells, a process called antigen presentation. Dendritic cells also release signals to activate and direct the immune response. Thus, dendritic cells act as a bridge between innate and adaptive immunity and help to identify and target pathogens and regulate the balance between immunogenicity and immune tolerance [53]. The differentiation of dendritic cells from immature to mature is associated with increased expression of 1-hydroxylase but decreased expression of VDR. Vitamin D enhances the production of anti-inflammatory cytokines by dendritic cells, which can help to regulate the immune response and prevent excessive inflammation [22]. Thus, in dendritic cells, 1,25(OH)_2_D_3_ can interfere with the differentiation and maturation process leading to a tolerogenic phenotype [54,55]. 

### 2.3. Vitamin D and Adaptive Immune System

Vitamin D is primarily an activator of innate immunity to augment the response to infection. However, it also regulates adaptive immunity. Adaptive immunity involves humoral immunity and cell-mediated immunity acting together against invading pathogens, a process leading to immunological memory after an initial encounter with a specific pathogen culminating in an enhanced response to future encounters with the pathogenic factor [56] via a fast and augmented production of neutralizing antibodies [57]. 

The effects of vitamin D on adaptive immunity involve its effects on Treg cells (Tregs) and on dendritic cells, which modulate T cell behavior. Tregs is a T cell subpopulation with immunosuppressive properties acting to maintain self-tolerance and prevent autoimmunity [58]. Vitamin D has been observed to promote the development and function of Tregs in vitro [59]. Vitamin D has direct and indirect effects on T cells [60]. T cells express the vitamin D receptor [61]. However, the optimal expression of vitamin D receptors in T cells is a late event [60]. Effector T cells are directly and indirectly affected, leading to a shift in the Th1/Th2 balance toward Th2 and a reduction in the Th17 response [59]. Once T cells are activated, 1,25(OH)_2_D_3_ inhibits IL-2 production [62]. The behavior of T cells is also indirectly modulated by vitamin D via its effects on dendritic cells. The vitamin D receptor is expressed at low levels in CD8+ and CD4+ T cells [61,63,64,65]. Following activation and addition of 1,25(OH)_2_D_3_, the expression of the vitamin D receptor is induced. Additionally, activated CD8+ cells produce 1-hydroxylase, which can convert 25(OH)D_3_ to the active 1,25(OH)_2_D_3_ [66,67]. Following T cell activation, 25(OH)D_3_ and 1,25(OH)_2_D_3_ inhibit T cell proliferation and cytokine production [61,64]. In the event of an infection, T cells are induced, which are important for clearing the pathogen. The effect of vitamin D is observed following the initiation of the T cell response to the infectious organism. In the infection models, T cells eliminate the pathogen, and the antigen is removed from the system. However, in an immune-mediated disease, the antigen persists, and T cells are chronically activated, producing inflammatory cytokines [68]. It has been suggested that vitamin D deficiency results in a reduced capacity to turn off T cells following activation [67]. 1,25(OH)_2_D_3_ was found to have anti-inflammatory effects on human macrophages [69]. 

B cells express immunoglobulin receptors in their plasma membrane recognizing antigenic epitopes. They produce autoantibodies and form B cell follicles. In activated B cells, the expression of vitamin D receptors and 1-hydroxylase is upregulated [70]. 1,25(OH)_2_D_3_ can induce apoptosis in B cells and can inhibit the formation and differentiation of B cells to plasma cells producing immunoglobulins [71]. 

All these actions are compatible with multiple and potent immunoregulatory actions of vitamin D, which may be related to it as a steroid hormone [72]. The local production and action within cells of the immune system are not under tight regulation by the parathyroid hormone or calcium levels and seem to be regulated by the amount of 25(OH)D_3_ in the circulation [73]. It appears that vitamin D possesses strong immunostimulatory properties and significant immunoregulatory properties, which may induce immune tolerance. Macrophages, monocytes, dendritic cells, Tregs, T cells, and B cells harbor VDR and vitamin D metabolizing enzymes. 

## 3. Vitamin D and Autoimmunity

Vitamin D has immunomodulatory properties [1,74,75], and during its discovery, it was shown to have immunostimulatory effects [76]. In the course of time, and as the autoimmune diseases were found to increase in prevalence [77], a worldwide prevalence of vitamin D deficiency was observed [19,78], suggesting a significant role of vitamin D in inducing immune tolerance [54,79,80] and a potential role of vitamin D deficiency in the development of autoimmune diseases [81,82,83] (Figure 1). Worldwide research provided evidence that vitamin D deficiency may contribute to the development of RA [24,84,85,86,87,88,89] and that it may be related to its activity and severity [24,88]. Vitamin D may have a sex-specific effect on autoimmunity as cross-talk between estrogen and vitamin D may exist [90]. Research also provided evidence that vitamin D deficiency may be related to systemic lupus erythematosus [26,91,92,93,94] and multiple sclerosis [95,96,97,98,99]. Vitamin D deficiency appears to be also highly prevalent in patients with inflammatory bowel disease [100] in relationship to disease activity [101]. Vitamin D acts to maintain the integrity of the intestinal barrier and is related to microbiota balance in these patients [102,103] and may contribute to the prevention of inflammatory bowel disease by supporting the integrity of the intestinal barrier, ensuring bacterial homeostasis and ameliorating disease progression via anti-inflammatory action [104]. Additionally, it may induce remission in patients with Crohn’s disease [105]. It has been postulated that vitamin D resistance may be observed in some patients suggesting that an individualized approach may be necessary for the treatment of vitamin D deficiency [106].

## 4. Rheumatoid Arthritis and Vitamin D

Vitamin D deficiency has been found to be a predisposing factor for the development of RA [89]. In a large study in female patients studied over a period of 11 years, a greater risk of RA was observed in patients with low vitamin D intake [107]. However, not all authors agree [108,109]. Song et al. [89] assessed in a meta-analysis the association between vitamin D intake and the risk of RA, and they found an association between vitamin D intake and RA incidence. Participants in the highest group for total vitamin D intake were found to have a 24.2% lower risk of developing RA as compared to those in the lowest group. 

Various studies from all over the world have shown that vitamin D deficiency is observed in patients with RA and seems to be inversely related to disease activity [23,88,110,111,112,113,114,115,116,117,118,119,120,121,122,123]. In a recent study in the Egyptian population, low vitamin D levels were observed in patients and controls. There was no difference in VDR polymorphisms between patients and controls [124]. In a meta-analysis of 24 studies published until 2015, Lin et al. [125] assessed the relationship between RA and vitamin D. They found significantly lower vitamin D levels in RA patients than in controls. They also observed an inverse relationship between vitamin D and disease activity, as assessed by DAS28. The inverse relationship between vitamin D and DAS28 was stronger in low-latitude countries and in developing as opposed to developed areas. An inverse relationship was also noted in their meta-analysis between vitamin D and CRP. Rheumatoid arthritis is a systemic inflammatory autoimmune disease [126,127]. Both T and B lymphocytes are involved in their pathogenesis [128,129,130]. Vitamin D may act on both T and B lymphocyte populations, thus enabling the regulation of the immune response necessary for the prevention or control of the disease. Higgins et al. [131] measured vitamin D levels in 176 patients with RA, and they found an inverse relationship between vitamin D levels and VAS, i.e., the patients’ rating of their symptoms on the visual analog score. Rheumatoid arthritis is a complex disease with various disease phenotypes. An effort was made to identify various disease phenotypes and to relate them to genotypes as they are expressed in gene profiling of synovial tissue samples [132]. Four major disease phenotypes of RA synovium were identified, namely lymphoid, myeloid, low inflammatory, and fibroid and a better response of the myeloid synovial disease phenotype to anti-TNFα treatment was observed. It seems possible that vitamin D deficiency may play a different role in the various RA disease phenotypes. 

## 5. Systemic Lupus Erythematosus and Vitamin D

Many studies over the years have evaluated the significance of vitamin D deficiency in SLE [26,91,93,133,134,135,136,137,138,139]. Various studies performed in countries all over the world have shown that vitamin D deficiency is observed in patients with SLE [26,93]. Many studies have shown that vitamin D deficiency may be associated with disease activity and renal involvement as well as with clinical characteristics. Following this observation, vitamin D was administered to patients with SLE; however, with conflicting results. 

In a study in a Spanish population, Ruiz-Irastorza et al. [140] found low 25(OH)D_3_ levels in SLE patients. In a seminal study, Amital et al. [26] measured 25(OH)D_3_ levels in 378 patients from Israeli and European cohorts, and they found that vitamin D levels were inversely related to disease activity. Vitamin D status was investigated by Shahin et al. [141] in a cohort of 57 treatment-naive SLE patients, and lower vitamin D levels were observed in SLE than in a group of controls. Low vitamin D levels were related to thrombocytopenia, and a negative correlation was observed between vitamin D and the inflammatory cytokines IL-17 and IL-23, as well as antinuclear antibodies. In a study of 290 Chinese patients with SLE, Mok et al. [93] investigated vitamin D levels and their relationship with disease activity. They found that 96% and 77% of their cohort had vitamin D insufficiency [25(OH)D_3_ levels < 30 ng/mL] and deficiency [25(OH)D_3_ < 15 ng/mL], respectively. An inverse correlation between 25(OH)D_3_ and disease activity as assessed by SLEDAI score (*p* = 0.003), and physicians’ global assessment (*p* = 0.003) was observed. Yao et al. [142] investigated bone mineral density and bone turnover markers in 80 patients with SLE, 85% premenopausal female patients, and they found lower 25(OH)D_3_ levels as compared to a control group (*p* < 0.001), which were inversely related with disease activity as assessed by SLEDAI (*p* < 0.001) and kidney involvement (*p* = 0.04). Kamen et al. [143] observed a trend toward lower vitamin D levels in their cohort compared to controls. Significantly low vitamin D levels were associated with renal disease and photosensitivity. Bogaczewicz et al. [144] found low vitamin D levels in a group of 49 SLE patients during the warm season and a trend towards low vitamin D levels during the cold season. Autoantibodies against 1,25(OH)_2_D_3_ were detected in three SLE patients. In a meta-analysis involving 18 studies with 1083 SLE patients, Bae and Lee [145] found significantly low vitamin D levels. In this meta-analysis, patients were stratified by ethnicity, and lower vitamin D levels were observed in European and Arab patients. In a meta-analysis performed in 2019 on 34 studies, lower vitamin D levels were observed in SLE patients than in controls [135]. Patients from areas in a latitude lower than 37°, either north or south of the equator, had lower vitamin D levels than the control populations. In a study performed in Greece at a latitude a little further north than 37° lower, 25(OH)D_3_ levels were found in SLE patients than in a control population. 

SLE patients may flare after sun exposure, and they are advised to avoid the sun as they have photosensitivity. Decreased vitamin D levels may be related to reduced sun exposure [146,147,148]. Corticosteroid administration may also be related to decreased vitamin D in SLE [149], although not all authors agree. Hydroxychloroquine administration in SLE may also be related to low vitamin D levels as it may affect vitamin D metabolism [149]. Autoantibodies against vitamin D may also be related to its low levels in SLE [144,150]. Low C_3_ and C_4_ levels may be related to low vitamin D levels in SLE, and both are related to increased disease activity [151,152]. Lupus nephritis may be related to low vitamin D [152,153]. In lupus and nephritis patients, VDR expression was studied with immunohistochemistry in renal biopsy specimens. The expression of VDR in the lupus nephritis group was lower and negatively related to disease activity [154,155]. Vitamin D may protect against aberrant podocyte autophagy in SLE nephritis [156]. 

Vitamin D receptor polymorphisms have been studied in patients with SLE [92,157,158]. Various vitamin D receptor polymorphisms have been shown to be related to SLE in various populations. However, data vary and have not shown any specific relationship between vitamin D receptor polymorphisms and the development of SLE in a population. 

It was further shown that vitamin D might contribute to the management of SLE [159,160,161,162,163,164,165]. Various dose schedules have been studied. Additionally, the daily dose of vitamin D needed in SLE may be higher than the dose needed for osteoporosis prevention and management. This relates to the fact that vitamin D metabolism in cells of the immune system is not under control by the parathyroid hormone. Therefore, vitamin D doses leading to serum levels of vitamin D > 30 ng/mL may be needed for vitamin D to exert its immunomodulating properties. 

## 6. Ankylosing Spondylitis and Vitamin D

Ankylosing spondylitis (AS) is an inflammatory autoimmune systemic rheumatic disorder that affects the enthesis. The disease is characterized by a bone-forming and a simultaneous bone resorbing process due to the effects of inflammatory cytokines [166]. Vitamin D levels have been studied in patients with ankylosing spondylitis. Ben-Shabat et al. [167] studied vitamin D levels in patients with AS in a large retrospective cohort study. They found vitamin D deficiency in patients with AS and a relationship between vitamin D deficiency and all-cause mortality in the patient group. Durmus et al. [168] measured 25(OH)D_3_ levels in a group of AS patients. They did not observe any significant difference in 25(OH)D_3_ levels between patients and controls. However, within the AS patient group, vitamin D levels were inversely related to pain, Bath AS Disease Activity Index (BASDAI), ESR, and CRP. In a study investigating the pathophysiology of vertebral fractures in AS patients, Lange et al. [169] observed a negative association between 1,25(OH)_2_D_3_ levels and disease activity. They concluded that bone metabolism and inflammatory activity might be closely related to AS. In a meta-analysis involving eight studies and a total of 533 AS patients, it was found that vitamin D may play a protective role against AS and that a reverse association exists between vitamin D levels and disease activity [170]. By contrast, an analysis investigating a possible causal role of vitamin D in AS concluded that vitamin D is not causally related to AS [171]. 

## 7. Psoriatic Arthritis and Vitamin D

PsA is a systemic autoimmune rheumatic disease affecting the skeleton and the skin. Vitamin D insufficiency has been observed with high prevalence in PsA patients [172]. However, no relationship between vitamin D insufficiency with disease activity was observed. In a study involving 72 patients with psoriasis and/or PsA, an inverse relationship was observed between vitamin D levels, the severity of skin involvement, and disease activity in PsA [173]. In a retrospective cross-sectional study involving 300 patients with plaque psoriasis with or without PsA performed in Brazil, a high prevalence of hypovitaminosis D was observed [174]. An inverse relationship was found between PASI, and vitamin D. Vitamin D was associated with the season and skin phototype. However, no favorable effect of oral vitamin D supplementation in patients with psoriasis could be verified [175].

## 8. Sjogren’s Syndrome and Vitamin D

Sjogren’s syndrome is an exocrinopathy affecting the salivary and lacrimar glands. It may occur as primary and as secondary in the context of another autoimmune rheumatic disorder, such as RA or SLE. Baldini et al. [176] measured vitamin D levels in a group of patients with primary Sjogren’s syndrome, and they found that vitamin D deficiency was associated with leukopenia. Erten et al. [177] measured vitamin D in a cohort of Sjogren’s syndrome patients, and they found lower vitamin D levels in the group of female patients as compared to the group of controls. In a study evaluating the effect of vitamin D deficiency on subclinical atherosclerosis in patients with primary Sjogren’s syndrome, an inverse relationship was observed between vitamin D deficiency, disease activity, and disease damage [178]. In a meta-analysis of studies on vitamin D levels and Sjogren’s syndrome published in a letter to the editor Li et al. [179] evaluated research performed on vitamin D and Sjogren’s syndrome up until 2019, and they found no difference between Sjogren’s syndrome and controls as far as vitamin D levels were concerned [179]. In a study involving 176 patients with primary Sjogren’s syndrome and 163 control subjects, low vitamin D levels were found to be related to the presence of peripheral neuropathy and lymphoma [180]. The authors concluded that vitamin D supplementation might be beneficial in Sjogren’s syndrome. 

## 9. Systemic Sclerosis and Vitamin D

Systemic sclerosis is a systemic autoimmune inflammatory fibrosing disorder [181,182,183,184]. It is characterized by skin, pulmonary, gastrointestinal, cardiac, and renal involvement [185]. The disease is severe and may lead to disability and death [186]. Systemic sclerosis is associated with vitamin D deficiency [187,188,189,190]. 

Data from in vitro studies have shown that vitamin D affects the fibrinogenic activity of fibroblasts by suppressing tissue growth factor β (TGFβ) [191,192]. In experimental models of scleroderma in mice, the application of vitamin D analogs has been shown to reduce fibrosis [193,194]. Additionally, vitamin D increased IL-10 production by Tregs derived from systemic sclerosis patients [193]. Low vitamin D levels have been observed in patients with systemic sclerosis [194,195,196,197,198,199,200,201,202,203]. These low vitamin D levels were irrespective of the season [204]. In a study by Arnson et al. [189], a negative correlation was observed between vitamin D and age. Most studies have not verified a relationship between vitamin D and disease activity in systemic sclerosis as opposed to Vacca et al. [205], who observed an inverse relationship between disease activity score and vitamin D. Carmel et al. [206] pointed out the possible relationship between low vitamin D levels and the presence of antibodies against vitamin D in systemic sclerosis. Two studies have noted lower vitamin D levels in patients with diffuse cutaneous systemic sclerosis as opposed to limited cutaneous systemic sclerosis [196,207]. Low vitamin D levels have been inversely related to the degree of cutaneous fibrosis, according to two important studies [191,197]. This has led to discussions on the possible effect of cutaneous involvement in systemic sclerosis being an obstacle in vitamin D synthesis [208]. Vitamin D deficiency has been observed to be related to digital ulcer presence in systemic sclerosis [187,209,210], although not all authors agree [31,200]. Pulmonary involvement is a common and serious manifestation of systemic sclerosis presenting as interstitial lung disease and pulmonary hypertension and has been related to vitamin D deficiency according to some [31,195,197,205] but not all studies [209]. Groseanu et al. [187] reported on the absence of lung fibrosis in the presence of normal vitamin D levels in patients with systemic sclerosis. Groseanu et al. [187] also reported an association between cardiac involvement and vitamin D deficiency in systemic sclerosis. Gastrointestinal involvement is one of the main manifestations of systemic sclerosis. However, vitamin D deficiency has not been found to be specifically correlated with it [196]. Renal disease is common in scleroderma, and scleroderma renal crisis may be life-threatening [211]. Renal involvement has been found to be related to vitamin D deficiency in systemic sclerosis by Trombetta et al. [31], while scleroderma renal crisis was found to be related to low vitamin D levels by Groseanu et al. [187]. Osteoporosis and hip fractures are commoner in systemic sclerosis patients than in healthy controls [194,212], and this appears to be related to vitamin D deficiency [213]. Vitamin D supplementation in systemic sclerosis may need to be performed with higher doses than in the healthy population as two studies have published results showing that relatively low doses may not be enough to raise vitamin D levels in this population [31,214], although this needs to be confirmed in larger studies. 

## 10. Inflammatory Myopathies and Vitamin D

Low serum levels of vitamin D have been observed in patients with idiopathic inflammatory myopathies, in particular polymyositis, dermatomyositis, inclusion body myositis, and juvenile dermatomyositis [215]. Low serum vitamin D levels in young patients with dermatomyositis may be a risk factor for the development of adult myositis. Low vitamin D levels in patients with inflammatory myopathies may contribute to the impaired bone health observed in this group of patients [216]. 

## 11. Vitamin D Supplementation

As there is strong evidence that vitamin D has immunomodulatory properties [1] and may prevent autoimmunity, vitamin D has been administered to patients with autoimmune diseases [86,159,217,218,219,220,221,222,223,224,225]. In particular, it has been administered to patients with RA (Table 1) and SLE (Table 2) with conflicting results. 

In an observational study, Di Franco et al. [226] evaluated patients with early RA and found that patients with low vitamin D levels at baseline had reduced responses to treatment as compared to those who had normal vitamin D levels at baseline. Based on this study and other studies showing low vitamin D levels in RA, Chandrashekara and Patted [223] administered cholecalciferol to a group of 73 patients with RA, high disease activity, and low vitamin D levels, and they observed a significant improvement in disease activity and vitamin D levels. Dehghan et al. [221] administered vitamin D or a placebo to RA patients with low vitamin D levels and evaluated the rate of flare and the decline in DAS28 in the two groups. They did not observe any difference in the rate of flare and in decline in disease activity between the patients who received or did not receive vitamin D supplementation. Kwon et al. [225] administered vitamin D at various daily doses for a year to patients with RA and osteoporosis who were on treatment with bisphosphonates and found a greater beneficial effect on bone mineral density in patients who received a dose of > 1000 IU daily than in those who received 800 IU daily. In an open-label randomized trial performed in India in patients with early RA, vitamin D deficiency was a risk factor for the development of active disease, vitamin D was inversely related to disease activity, and vitamin D supplementation resulted in greater pain relief [224]. In a meta-analysis of six studies published between 2011 and 2018 involving 438 patients with RA, it was reported that vitamin D supplementation improved DAS28, tender joint count, and ESR [227], while in some subgroups, vitamin D supplementation also improved VAS. In a randomized controlled trial involving 25,871 participants, Hahn et al. [21] administered vitamin D 2000 IU/day or placebo and marine-derived omega 3 fatty acids and estimated the incidence of autoimmune diseases over a period of observation of 5.3 years. They found that vitamin D supplementation with or without omega 3 fatty acids decreased autoimmune disease by 22%, while omega 3 fatty acid supplementation with or without vitamin D decreased autoimmune disease by 15%. In another study, vitamin D supplementation as compared to placebo in patients with RA on treatment with methotrexate did not show any significant benefit of vitamin D over placebo [220]. In a meta-analysis involving five studies in RA vitamin D supplementation decreased disease recurrence [228]. It appears that vitamin D supplementation may have beneficial effects on RA. However, more studies are needed on the subject. 

Vitamin D deficiency has been noted in SLE patients in many studies worldwide and has been found to be related to disease activity. As vitamin D has immunomodulatory properties and may mitigate autoimmunity, vitamin D supplementation has been performed in SLE patients. In particular, vitamin D was administered to patients with SLE who had either vitamin D deficiency or insufficiency for 1 year. Vitamin D significantly improved disease activity as assessed by SLEDAI-2K and fatigue and significantly decreased anti-dsDNA, and insignificantly suppressed interferon signature gene expression [164]. In a study performed in India, north and south, Kavadinchanda et al. [165] assessed vitamin D levels in SLE patients from the north and south of India, and they found lower vitamin D levels in patients from the north as compared to those from the south of India. They performed supplementation with either routine or high-dose vitamin D in SLE patients. They did not observe any difference in the number of flares between the groups. Al-Kushi et al. [163] supplemented a group of 81 SLE patients with vitamin D and calcium. All patients had low vitamin D levels at baseline. They did not observe any effect of vitamin D supplementation on immune markers or disease activity, but they observed improvement in osteopenia and osteoporosis. Andreoli et al. [159] supplemented 34 female SLE patients with cholecalciferol either with a standard or an intensive regimen. Vitamin D levels increased. However, no effect was observed in disease activity or serology. Aranow et al. [160] administered vitamin D 2000 IU or 4000 IU daily for 12 weeks in 57 SLE patients, and they assessed interferon signatures. They did not find any effect of vitamin D on interferon signature. Marinho et al. [162] evaluated the effect of vitamin D supplementation in SLE patients on the FoxP3+/IL-17A ratio as an index of immune function. They found increased vitamin D levels and an improved FoxP3+/IL-17A ratio. Singgih Wahono et al. [229] examined the combined effect of vitamin D and curcumin in SLE patients, and they did not observe any effect of the combined treatment on disease activity and inflammatory cytokines. In conclusion, vitamin D supplementation may have beneficial effects on SLE. However, the results of the literature so far are controversial, and more research is needed on the subject. 

## 12. Vitamin D Deficiency in Autoimmune Rheumatic Diseases

Vitamin D deficiency may be a factor contributing to the development of autoimmune rheumatic diseases. Vitamin D is an immunomodulating factor and may modulate the immune response. Thus, vitamin D induces a tolerogenic phenotype by acting on antigen-presenting cells, monocytes, and natural killer cells [20]. In particular, vitamin D induces a decrease in the expression of major histocompatibility complex II (MHC) II and co-stimulatory molecules in antigen-presenting cells [54], thus leading to decreased antigen presentation, production of interleukin-12 and increased interleukin-10. Vitamin D also seems to suppress the expression of toll-like receptors and the production of inflammatory cytokines by monocytes [230]. The vitamin also modulates the function of natural killer cells [231]. The vitamin is produced locally by monocytes acting in an intracrine fashion [232], resulting in an immunity shift from an inflammatory to a tolerogenic state. Vitamin D promotes a shift from a Th1 to Th2 immune profile [233] and promotes the differentiation of Tregs [234]. Thus, its deficiency may be related to the development of autoimmunity [235]. Alternatively, low vitamin D levels may be related to the presence of inflammation in autoimmune rheumatic diseases [236], a status known as reverse causation [133], as vitamin D may be a reverse index of inflammation [237]. 

Vitamin D deficiency is a modern pandemic [238]. It affects the worldwide population, but it may be particularly prevalent amongst the population used to cover most of their body with clothes [239] or having a different skin pigmentation [240] or the population residing most of the time within buildings, thus avoiding the effects of sun exposure [241]. The optimal level of vitamin D, 25(OH)D_3_, for health, is not known. However, data from our ancestors who lived in nature and were exposed to the sun as they were hunters/gatherers are that they maintained a level of 10–50 ng/mL. More recent data on populations living a primitive way of life is that they have a reported level of 25(OH)D_3_ of 40–60 ng/m [242]. Moreover, the optimal level for the prevention or mitigation of an autoimmune process is not known and may be higher than that reported for general health outcomes [23,133,243]. In addition, the optimal method of vitamin D supplementation in patients with autoimmune diseases is still under research. 

## 13. Conclusions

Vitamin D has immunomodulatory properties, and its deficiency may be related to the development of autoimmune diseases such as RA and SLE. Data on the relationship between other autoimmune rheumatic diseases such as AS, PsA, Sjogren’s syndrome, systemic sclerosis, and idiopathic inflammatory myopathies have also been published, corroborating findings from the original studies on RA and SLE. Data on vitamin D supplementation have shown that vitamin D may be beneficial in autoimmune rheumatic diseases and may mitigate the autoimmune process and improve pain in the context of these diseases. 

## Figures and Tables

**Figure 1 biomolecules-13-00709-f001:**
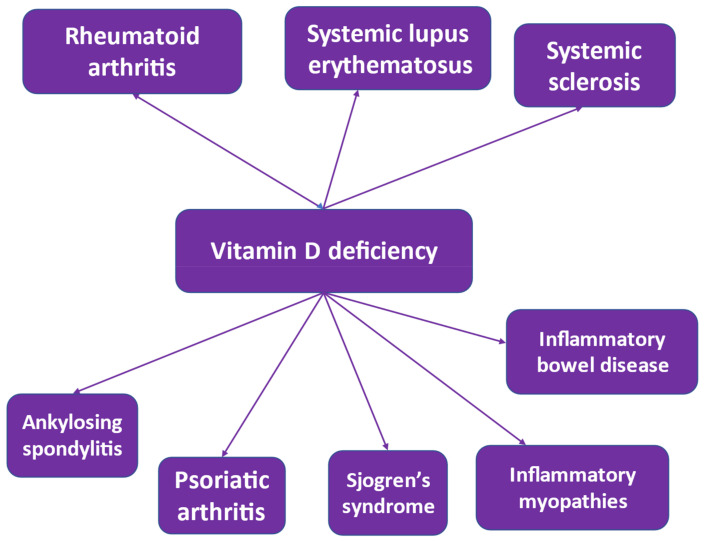
Autoimmune rheumatic diseases related to vitamin D deficiency.

**Table 1 biomolecules-13-00709-t001:** Trials in which vitamin D was administered to rheumatoid arthritis (RA) patients, RCT = randomized controlled trial.

Authors (Date)	Type of Study	Study Population	Results
Andjelkovic et al. (1999) [218]	Open label	RA = 19	Improvement of clinical indices
Gopinath and Danda (2011) [219]	Open label	RA = 121	Improved pain relief
Salesi et al. (2012) [220]	Double-blind	RA = 117	Improved DAS28
Dheghan et al. (2014) [221]	RCT	RA = 80	No improvement in relapse rate
Hansen et al. (2014) [222]	RCT	RA = 22	Improved bone formation
Buondonno et al. (2017) [86]	RCT	RA = 70	Reduced inflammatory cytokines
Chandrashekara and Patted (2017) [223]	Open label	RA = 73	Improved DAS28, CRP
Mukherjee et al. (2019) [224]	Open label	RA = 25	Improved pain
Kwon et al. (2020) [225]	Open label	RA with osteoporosis = 187	Improved bone mineral density

**Table 2 biomolecules-13-00709-t002:** Trials in which vitamin D was administered to systemic lupus erythematosus (SLE) patients, RCT = randomized controlled trial.

Authors (Date)	Type of Study	Study Population	Results
Andreoli et al. (2015) [159]	Randomized prospective study	SLE = 34 female	No significant effect on clinical outcomes
Aranow et al. (2015) [160]	Double-blind placebo controlled	SLE = 54	No significant effects
Karimzadeh et al. (2017) [161]	Double-blind placebo controlled	SLE = 90	No significant effects
Marinho et al. (2017) [162]	Prospective cross-sectional study	SLE = 24	Decreased SLEDAI, immunologic effects
Al-Kushi et al. (2018) [163]	RCT	SLE = 81	Improved osteopenia osteoporosis
Magro et al. (2021) [164]	Open label	SLE = 31	Improved disease activity, fatique
Kavadichanda et al. (2023) [165]	Cross sectional study	SLE = 702	No significant effects

## Data Availability

Not applicable.

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
