# Peer review of "Vitamin D and Autoimmune Rheumatic Diseases"

_biomolecules, 2023, doi:10.3390/biom13040709_

Round 1
Reviewer 1 Report
In the present review, the authors write about vitamin D deficiency and autoimmune diseases, while the title is vitamin D and rheumatic diseases. I think the title didn’t explain the review idea. The structure of the review and the arrangement of the information flow to attract the reader's attention were very poor. There are several grammar mistakes as well as missing structure in several sentences. I think the authors must rewrite that review after extensive reading of the references. Several pionts needs revision as follows:
- In the abstract:
- The authors write "Vitamin D is a hormone", This is wrong information, and it is written in the introduction. I think the authors didn’t read references 1 and 2, as they established that the metabolites of vitamin D in the body are steroid hormones. Please read the references carefully.
- In the introduction:
- The introduction lacks specific information that is necessary to explain the idea of the review and the studied items.
- The introduction lacks the gap that the review resolved and the novelty of that review compared to other similar reviews.
- The references are too old; please add new references.
- In vitamin D and innate immunity:
- Change 1a-hydroxylase into 1-hydroxylase in all the manuscripts.
- In line number 58 "vitamin D receptor VDR", please add VDR inside brackets.
4. Vitamin D and adaptive immune system:
o In line 117, "T cells express the vitamin D receptor60", what does the number 60 refer to? Is it a reference?
o In lines number 134 – 135, "In a previous study, 1,25(OH)2D3 was observed to diminish the proliferation of peripheral blood mononuclear cells and the production of inflammatory cytokines after stimulation with T-cell specific mitogens [68]", while the reference number 68 is from 1987.
o In line number 140, "1,25(OH)2D3 can induce apoptosis in B cells, can inhibit memory B cell formation and may prevent the differentiation of B cells to plasma cells producing immunoglobulins", please revise this sentence with the next reference.
o Figure 1 didn’t explain any new information for the reader; I think it must be deleted and replaced by one sentence at the end of the paragraph.
5. In Vitamin D and autoimmunity:
o " In the course of time, and as the autoimmune diseases were found to increase in prevalence76 a worldwide prevalence of vitamin D deficiency was observed", What does 76 refers to? And another numbers like 100, … etc; Please revise.
o Figure 2 didn’t explain any new information to the reader, so, it must be deleted and replaced by one sentence in the paragraph.
6. In Rheumatoid arthritis and vitamin D:
o Paragraphs in line numbers 196-201 and 204 – 205 must be transformed after the title as first the RA definition, then the correlation between vitamin D and RA.
7. In line number 267, what does PTH refer to?
8. The conclusion didn’t clarify the idea of the review; it must be improved.
Author Response
We thank the reviewer for his efforts to review the paper. Please see the attachment.

Reviewer 2 Report
1) Abstract. Vitamin D deficiency has also been observed in systemic sclerosis. Vitamin D deficiency may be implicated in the pathogenesis of autoimmunity. Alternatively, vitamin D may be a reverse marker of inflammation. Vitamin D may be administered to prevent autoimmune rheumatic disorders and may be used as supplementary treatment in patients with autoimmmune rheumatic disorders as it may reduce pain. Please, improve thi paragraph.
2) Introduction. L38-42. Vitamin D defi- ciency has been observed in various rheumatic autoimmune diseases and it may be re- lated to the pathogenesis of autoimmunity [16,18]. Also, vitamin D is administered in patients with rheumatic autoimmune diseases to control pain and prevent disease exac- erbation [19-21]. Vitamin D has been proposed to prevent autoimmunity [21]. In order to discuss the previously described points, important references are needed to be added, such as:
a- Vitamin D deficiency and clinical correlations in systemic sclerosis patients: A retrospective analysis for possible future developments. PLoS One. 2017 Jun 9;12(6):e0179062. doi: 10.1371/journal.pone.0179062.
b- Trabecular Bone Score and Bone Quality in Systemic Lupus Erythematosus Patients. Front Med (Lausanne). 2020 Sep 30;7:574842. doi: 10.3389/fmed.2020.574842.
c- The Effects of Vitamin D on Immune System and Inflammatory Diseases. Biomolecules 2021, 11, 1624. https://doi.org/10.3390/biom11111624.
3) Introdusction. L47-49. In particular, the relationship of vit- amin D deficiency with autoimmune rheumatic diseases has attracted the attention of the scientific community and has led to various studies of vitamin D supplementation to mit igate the severity of autoimmune rheumatic diseases [21]. Please improve the description of study aim.
4) Figure 1. Cells of the immune system expressing the vitamin D receptor (VDR) and vitamin D me- 150 tabolizing enzymes. Please, improve this figure and add an image of different cells.
5) Figure 2. Please, improve
6) 11. Conclusion L442-448. Vitamin D has immunomodulatory properties and its deficiency may be related to the development of autoimmune diseases such as RA and SLE. Data on the relationship between other autoimmune rheumatic diseases such as AS, systemic sclerosis and Sjogren’s syndrome have also been published thus lending more weight to the original data on RA and SLE. However, data on vitamin D supplementation have not shown consistent results and indicate that more research may be needed on the relationship be- tween vitamin D and autoimmunity. Please improve the conclusions and underline the clinical implications of the study.
Author Response

(The authors gave the same response as above.)

Reviewer 3 Report
The authors extensively described all potential role of vitamin D in rheumatic diseases. The work should be reviewed in some points:
- A section on methods is missing. The authors should specify the research criteria, the source of the research, the period, etc.
- Some rheumatologic diseases, such as spondylarthritis and psoriatic arthritis, inflammatory myositis and other connective tissues diseases were not considered. The authors should add considerations on these pathologies and the potential role of vitamin D in them.
- There are no critical views of the authors. Author’s considerations should be added.
- The figure 1 should be enriched with more details.
- The role of vitamin D on the immune system should be explained in more detail adding another figure
- A schematic table on potential role of vitamin D in different rheumatic diseases should be added.
Some writing inaccuracies are present in the text:
- Line 422: “Vitamin also modulates the function of natural” – Should be added “D”
After these corrections the work could be reconsidered for publication
Author Response

(The authors gave the same response as above.)

Reviewer 4 Report
Although the review is of an interesting topic, I have some concerns about the way in which it has been written. A lot of sentences explain the same concept throughout the manuscript for several times making it redundant in its form and, at the same time, avoinging to give important findings concerning the mechanism of action.
A review that it is easy to understand cannot be written in a simplistic way. In my opinion more than half of the manuscript does not give the reader an explanation of the written concepts.
The first three paragraphs are a little bit confusing about the well-known properties of vitamin D. Please, the authors should summarize them in a better way.
Figures 1 and 2 do not contain self-explain meanings. The authors should remove them. Why the authors have not insert Crohn’s disease in Figure 2?
Note that abbreviations can be insert for the first time next to the complete name (lane 159 - development of RA and so on).
The authors should correct the mistakes about the presence of number in the sentence as in the following ones for example:
- In Lane 163 - Vitamin D deficiency appears to be also highly prevalent in patients with inflammatory bowel disease100 in relationship to disease activity [101] - probably there is a mistake.
- In lane 348-9 - As there is strong evidence that vitamin D has immunomodulatory properties1 and may prevent autoimmunity, vitamin D has been administered to patients with autoimmune diseases.
What are higher doses about the concept in lane 267 - Therefore, higher doses may be needed for vitamin D to exert its immunomodulating properties. Moreover, when they indicated "low vitamin D levels" the authors should be more precise and give a range when human studies are mentioned.
Lane 142 - The sentence "All these actions are compatible with a multiple and potent immunoregulatory action of vitamin D (figure 1), which is also related to its molecule being a steroid [71]" should be explained.
Lane 252 - "Renal involvement has been found to
be related with low vitamin D in various studies [152,153] " should be explained.
Lane 160 - A cross-talk between estrogen and vitamin D has been postulated implying a sex-specific effect of vitamin D in autoimmunity [89] should be explained.
6. Vitamin D and ankylosing spondylitis. Why not 6. Ankylosing spondylitis Vitamin D as title?
I suggest to the authors a huge revision of each sentence in its content trying to insert the meaning of the clinical data proposed on the basis of the known molecular mechanisms, often already given in the cited publications.
Abstract, paragraphs 9, 10 and 11 are fine as well as table 1 and 2 (insert abbreviations for table 1 and 2 to make them understandable).
Please, remove all the redundant sentences and performe an extensive editing of English language and style.
Author Response

(The authors gave the same response as above.)

Round 2
Reviewer 1 Report
Authors must refer to Figure 1 in the text.
Author Response
We wish to thank the reviewer for his efforts to review the paper.
Figure 1 has been referred to within the text in line 201.
Reviewer 3 Report
The authors revised the article as requestedAuthor Response
We wish to thank the reviewer for his efforts to review the paper.
Reviewer 4 Report
In my opinion the review has been improved and in this form is suitable to be published.
I suggest to redraw figure 1 making it more homogeneous in the form of the boxes and indicators. Probably also using a different colour for the boxes as blue and green are always in relation to health and not disease.
Author Response
We wish to thank the reviewer for his efforts to review and improve the paper.
The figure has been redrawn and the color has been altered as suggested by the reviewer.